# Non-Hermitian Hamiltonians and Quantum Transport in Multi-Terminal Conductors

**DOI:** 10.3390/e22040459

**Published:** 2020-04-17

**Authors:** Nikolay M. Shubin, Alexander A. Gorbatsevich, Gennadiy Ya. Krasnikov

**Affiliations:** 1P.N. Lebedev Physical Institute of the Russian Academy of Sciences, Moscow 119991, Russia; shubinnm@lebedev.ru; 2JSC Molecular Electronics Research Institute, Zelenograd, Moscow 124460, Russia; gkrasnikov@niime.ru

**Keywords:** non-Hermitian Hamiltonians, open quantum systems, resonances, quantum conductor, quantum interference

## Abstract

We study the transport properties of multi-terminal Hermitian structures within the non-equilibrium Green’s function formalism in a tight-binding approximation. We show that non-Hermitian Hamiltonians naturally appear in the description of coherent tunneling and are indispensable for the derivation of a general compact expression for the lead-to-lead transmission coefficients of an arbitrary multi-terminal system. This expression can be easily analyzed, and a robust set of conditions for finding zero and unity transmissions (even in the presence of extra electrodes) can be formulated. Using the proposed formalism, a detailed comparison between three- and two-terminal systems is performed, and it is shown, in particular, that transmission at bound states in the continuum does not change with the third electrode insertion. The main conclusions are illustratively exemplified by some three-terminal toy models. For instance, the influence of the tunneling coupling to the gate electrode is discussed for a model of quantum interference transistor. The results of this paper will be of high interest, in particular, within the field of quantum design of molecular electronic devices.

## 1. Introduction

Traditional treatment of quantum transport is based on the scattering theory [1]. A correspondence between the scattering matrix (*S*-matrix) and Hamiltonian approaches is established within the framework of Fano–Feshbach formalism [2,3,4]. In this formalism, an effective non-Hermitian Hamiltonian is introduced, whose complex eigenvalues coincide with scattering matrix poles. Non-Hermitian Hamiltonians are of great interest in modern quantum physics, as they can describe various phenomena beyond the traditional paradigm of Hermitian operators in a very robust and illustrative way [5]. Non-Hermitian Hamiltonians typically appear in the study of open quantum systems (OQS), where the total Hermitian Hamiltonian of the whole system is projected on the states of its subsystem of interest [2] resulting in a non-Hermitian effective Hamiltonian. OQS being a part of a bigger system, does not have stationary eigenstates. Eigenstates of the projected effective Hamiltonian are called resonant states, and corresponding eigenvalues are complex, with the real part indicating the energy and the imaginary part showing the decay rate (outgoing momentum flux [6]). However, incoming and outgoing (scattered) waves are characterized by real energies. Hence, the connection between complex eigenvalues of an effective Hamiltonian (poles of *S*-matrix) with real energies of transmission peaks/dips is of high importance. Usually, one associates energies of tunneling transmission resonances with real parts of the *S*-matrix poles. This interpretation is adequate only in the case of well-separated and narrow resonances. If perfect (unity-valued) resonances become wider and closer to each other, they can coalesce, resulting in a single transmission peak with amplitude smaller than unity [7]. This phenomenon cannot be detected from the analysis of the *S*-matrix poles alone [8]. In complex systems, where destructive quantum interference (DQI) is possible, much more complicated interference phenomena are expected, so the traditional *S*-matrix (or effective Hamiltonian) point of view cannot handle all the variety of possible interference effects in quantum transport.

Recently, it has been shown that a stationary scattering problem within two channels (two terminals) can be regarded from a different point of view, where some new non-Hermitian Hamiltonian plays the role [9,10,11]. This new auxiliary non-Hermitian Hamiltonian turned to be PT-symmetric in spatially symmetric systems [9,11]. Here, P stands for space inversion and T for time reversal operations. It is known that such Hamiltonians have eigenvalues, which in general, are complex conjugate to each other and can be real [12,13]. This is impossible for effective Hamiltonian as its eigenvalues (*S*-matrix poles) are located in the lower half of a complex energy plane. In our previous works [8,14,15], we have thoroughly studied PT-symmetric two-terminal quantum conductors and have established a direct correspondence between perfect transmission peaks and real eigenvalues of this non-Hermitian auxiliary Hamiltonian. Within this approach, resonance coalescence can be described straightforwardly as a PT-symmetry breaking of the auxiliary Hamiltonian at its exceptional point (EP) [16], where two real eigenvalues coalesce and turn into a complex conjugate pair. Moreover, DQI and formation of bound states in the continuum (BIC) [17] can also be described using our technique.

Physical properties of multi-terminal conductors are significantly richer than those of two-terminal structures [18,19,20]. The scattering matrix approach for studying quantum transport has been generalized to the description of multi-terminal conductors by Büttiker [21,22]. In particular, he has shown that the insertion of extra electrodes can be considered as the emergence of additional inelastic scattering channels, which results in dephasing [23]. It also destroys the perfect transparency of the two-terminal quantum conductor at resonance. In the present paper, we propose a theory of quantum transport in multi-terminal conductors, which generalizes the results of [15].

Using the developed formalism, we show the possibility of perfect transmission in three-terminal configurations and present simple rules of how to design multi-terminal quantum conductors with perfect transparency. Additionally, correspondence between three- and two-terminal configurations of structures possessing BICs is discussed. The paper is organized as follows. In Section 2, we describe the model of a quantum conductor and state some standard formulas for the transmission coefficient calculation using the effective Hamiltonian approach. In Section 3, one can find the generalization of the auxiliary Hamiltonian approach to the case of multi-terminal conductors. Properties of derived transmission coefficients and conditions for perfect and zero transparency are discussed. Section 4 provides illustrative examples of three-terminal systems, including a model of quantum interference transistor. In Section 5, we show correspondence between two- and three-terminal systems and discuss transmission at BICs. Finally, there is a summary in Section 6.

## 2. Multi-Terminal Quantum Conductor

We consider an arbitrary *N*-site structure (a molecule or a quantum dot array) connected to *M* semi-infinite leads. Each site has a single localized state with energy $\varepsilon_{i}$. The full Hamiltonian of this system within the tight-binding approximation is the following
(1)$$\hat{H}={\hat{H}}_{0}+{\hat{H}}_{1}+\ldots +{\hat{H}}_{M}+{\hat{H}}_{int}^{1}+\ldots +{\hat{H}}_{int}^{M}.$$

The first term in Equation (Equation 1) is the bare Hamiltonian of the *N*-site structure:(2)$${\hat{H}}_{0}=\sum_{i=1}^{N}\varepsilon_{i}a_{i}^{†}a_{i}+\sum_{i,j=1,i<j}^{N}\left(\tau_{ij}a_{j}^{†}a_{i}+h.c.\right),$$
where $a_{i}^{†}\left(a_{i}\right)$ is the creation (annihilation) operator of the electron on the *i*-th site and $\tau_{ij}$ is the hopping integral between the *i*-th and the *j*-th sites.

The α-th lead with the energy spectrum $\varepsilon_{lead}^{\alpha }=\varepsilon_{lead}^{\alpha }\left(p\right)$ is described by the Hamiltonian ${\hat{H}}_{\alpha }$:(3)$${\hat{H}}_{\alpha }=\sum_{p}\varepsilon_{lead}^{\alpha }\left(p\right)a_{p}^{\alpha †}a_{p}^{\alpha }.$$

Operator $a_{p}^{\alpha }$ in Equation (Equation 3) corresponds to the state in the α-th lead with momentum *p*. Term ${\hat{H}}_{int}^{\alpha }$ in Equation (Equation 1) describes the coupling between the state with momentum *p* in the α-th lead and the *i*-th site of the structure for all *p* and *i*:(4)$${\hat{H}}_{int}^{\alpha }=\sum_{p,i}\left(\gamma_{p,i}^{\alpha }a_{i}^{†}a_{p}^{\alpha }+h.c.\right).$$

In general, matrix elements $\gamma_{p,i}^{\alpha }$ depend on energy and momentum.

Transmission probability from the lead α to the lead β (α,β∈{1,…,M}) is given by the standard expression [1]:(5)$$T_{\alpha \beta }=4\mathrm{Tr}\;\left({\hat{\Gamma }}^{\beta }{\hat{G}}^{r}{\hat{\Gamma }}^{\alpha }{\hat{G}}^{a}\right).$$

Here ${\hat{G}}^{r}$ and ${\hat{G}}^{a}={\left({\hat{G}}^{r}\right)}^{†}$ are correspondingly retarded and advanced Green’s functions of the system:(6)$${\hat{G}}^{r}={\left(E\hat{I}-{\hat{H}}_{eff}\right)}^{-1},$$
where $\hat{I}$ is the N×N identity matrix and ${\hat{H}}_{eff}$ is the effective Hamiltonian [2] of the system:(7)$${\hat{H}}_{eff}={\hat{H}}_{0}+{\hat{\Sigma }}^{1}+\ldots +{\hat{\Sigma }}^{M}.$$

Here ${\hat{\Sigma }}^{\alpha }$ is the self-energy of the α-th lead. The Hermitian matrix ${\hat{\Gamma }}^{\alpha }$ from Equation (Equation 5) is the anti-Hermitian part of the corresponding lead self-energy:(8)$${\hat{\Sigma }}^{\alpha }={\hat{\delta }}^{\alpha }-i{\hat{\Gamma }}^{\alpha }.$$

For semi-infinite single-channel leads one can derive self-energy in the tight-binding approximation as follows [24]:(9)$$\Sigma_{ij}^{\alpha }=\sum_{p,p^{\prime }}\gamma_{p,i}^{\alpha }{\left({\hat{G}}_{\alpha }^{r}\right)}_{pp^{\prime }}\gamma_{p^{\prime },j}^{\alpha *}$$
where ${\hat{G}}_{\alpha }^{r}$ is the retarded Green’s function of the isolated α-th lead, which is diagonal in the basis of momentum eigenfucntions:(10)$${\left({\hat{G}}_{\alpha }^{r}\right)}_{pp^{\prime }}={\left[{\left(E-{\hat{H}}_{\alpha }\right)}^{-1}\right]}_{pp^{\prime }}={\left[E-\varepsilon_{lead}^{\alpha }\left(p\right)+i0\right]}^{-1}\delta_{pp^{\prime }}.$$

Assuming that the matrix elements $\gamma_{p,i}^{\alpha }=\gamma_{i}^{\alpha }\left(\varepsilon_{lead}^{\alpha }\right)$ depend on the energy $\varepsilon_{lead}^{\alpha }=\varepsilon_{lead}^{\alpha }\left(p\right)$ but not on the momentum *p*, Hermitian and anti-Hermitian parts of the α-th lead self-energy can be written as follows:(11)$$\begin{array}{c} \delta_{ij}^{\alpha }\left(E\right)=p.v.\int \frac{\gamma_{i}^{\alpha }\left(E^{\prime }\right)\gamma_{j}^{\alpha *}\left(E^{\prime }\right)\rho_{\alpha }\left(E^{\prime }\right)}{E-E^{\prime }}dE^{\prime },\\ \Gamma_{ij}^{\alpha }\left(E\right)=\pi \gamma_{i}^{\alpha }\left(E\right)\gamma_{j}^{\alpha *}\left(E\right)\rho_{\alpha }\left(E\right). \end{array}$$

Here $\rho_{\alpha }$ is the density of states for the α-th lead. Thus, the transmission coefficient $T_{\alpha \beta }$ becomes
(12)$$T_{\alpha \beta }=\frac{4\sum_{i,j,m,k=1}^{N}{\left(-1\right)}^{i+j+m+k}M_{ij}^{*}M_{mk}\Gamma_{jk}^{\beta }\Gamma_{mi}^{\alpha }}{{\left|\det\left(E\hat{I}-{\hat{H}}_{eff}\right)\right|}^{2}},$$
where $M_{ij}$ are the minors of the $\left(E\hat{I}-{\hat{H}}_{eff}\right)$ matrix.

## 3. Transmission Coefficient in Multi-Terminal Quantum Conductor

### 3.1. Formula for Transmission Coefficient

Using Equation (Equation 11) and the conventional approach to the description of decays (see, e.g., Ref. [25]), matrix ${\hat{\Gamma }}^{\alpha }$ can be written as:(13)$${\hat{\Gamma }}^{\alpha }=u_{\alpha }u_{\alpha }^{†},$$
with $u_{\alpha ,i}=\sqrt{\pi \rho_{\alpha }}\gamma_{i}^{\alpha }$ being the *i*-th element of the column-vector $u_{\alpha }$. Using Equation (Equation 13) we can rewrite Equation (Equation 5) in a new form, different from Equation (Equation 12), which enables one to provide clear analysis of various interference phenomena. For brevity, we introduce a matrix
(14)$${\hat{A}}^{\alpha \beta }={\hat{A}}_{1}+i{\hat{A}}_{2}^{\alpha \beta },$$
where
(15)$${\hat{A}}_{1}=E\hat{I}-{\hat{H}}_{0}-\sum_{\sigma =1}^{M}{\hat{\delta }}^{\sigma },\;A_{2}^{\alpha \beta }=\sum_{\begin{array}{c} \sigma =1,\\ \sigma \neq \alpha ,\beta \end{array}}^{M}{\hat{\Gamma }}^{\sigma }.$$

The matrix ${\hat{A}}^{\alpha \beta }$ is non-Hermitian and Hermitian matrices ${\hat{A}}_{1}$ and ${\hat{A}}_{2}^{\alpha \beta }$ represent its Hermitian and anti-Hermitian parts respectively. It should be noted that Hermitian part ${\hat{A}}_{1}$ is independent of a particular choice of α and β. The effective Hamiltonian (Equation 7) in this notation can be written as
(16)$${\hat{H}}_{eff}=E\hat{I}-{\hat{A}}^{\alpha \beta }-i{\hat{\Gamma }}^{\alpha }-i{\hat{\Gamma }}^{\beta }=E\hat{I}-{\hat{A}}^{\alpha \beta }-iu_{\alpha }u_{\alpha }^{†}-iu_{\beta }u_{\beta }^{†}.$$

Non-Hermiticity of the matrix ${\hat{A}}^{\alpha \beta }$ is the key difference between the case of multi-terminal structures and two-terminal structures considered in Ref. [15]. Using ${\hat{A}}^{\alpha \beta }$ from Equation (Equation 14) one can get for the transmission coefficient:(17)$$\begin{array}{c} T_{\alpha \beta }=4\mathrm{Tr}\;\left\{u_{\beta }u_{\beta }^{†}{\left({\hat{A}}^{\alpha \beta }+iu_{\alpha }u_{\alpha }^{†}+iu_{\beta }u_{\beta }^{†}\right)}^{-1}u_{\alpha }u_{\alpha }^{†}{\left[{\left({\hat{A}}^{\alpha \beta }+iu_{\alpha }u_{\alpha }^{†}+iu_{\beta }u_{\beta }^{†}\right)}^{-1}\right]}^{†}\right\}\\ \;\;=4{\left|u_{\beta }^{†}{\left({\hat{A}}^{\alpha \beta }+iu_{\alpha }u_{\alpha }^{†}+iu_{\beta }u_{\beta }^{†}\right)}^{-1}u_{\alpha }\right|}^{2}. \end{array}$$

Utilizing the Sherman-Morrison formula [26] and matrix determinant lemma [27] to Equation (Equation 17) we can derive the following:(18)$$T_{\alpha \beta }=\frac{4{\left|\det{\hat{A}}^{\alpha \beta }\right|}^{2}{\left|u_{\beta }^{†}{\left({\hat{A}}^{\alpha \beta }\right)}^{-1}u_{\alpha }\right|}^{2}}{{\left|\det\left({\hat{A}}^{\alpha \beta }+iu_{\alpha }u_{\alpha }^{†}+iu_{\beta }u_{\beta }^{†}\right)\right|}^{2}}.$$

According to the definitions in Equations (Equation 13), (Equation 14) and (Equation 16) the denominator of Equation (Equation 18) is nothing more than the characteristic determinant of the effective Hamiltonian. From Equation (Equation 18) it follows that the numerator of the transmission coefficient is a square module of a certain energy-dependent quantity $P_{0}^{\alpha \beta }$, which is defined up to an arbitrary phase factor:(19)$$P_{0}^{\alpha \beta }=2u_{\beta }^{†}\left(\mathrm{adj}\;{\hat{A}}^{\alpha \beta }\right)u_{\alpha }.$$

Here $\mathrm{adj}\;{\hat{A}}^{\alpha \beta }$ is the adjugate matrix of ${\hat{A}}^{\alpha \beta }$.

Getting apart the term $4|\det{\hat{A}}^{\alpha \beta }{|}^{2}|u_{\beta }^{†}{\left({\hat{A}}^{\alpha \beta }\right)}^{-1}u_{\alpha }{|}^{2}={\left|P_{0}^{\alpha \beta }\right|}^{2}$ in the denominator of Equation (Equation 18) and simplifying the rest terms by the matrix determinant lemma, one can figure out that
(20)$${\left|\det\left(E\hat{I}-{\hat{H}}_{eff}\right)\right|}^{2}={\left|\det\left({\hat{A}}^{\alpha \beta }+i{\hat{\Gamma }}^{\alpha }+i{\hat{\Gamma }}^{\beta }\right)\right|}^{2}={\left|P_{0}^{\alpha \beta }\right|}^{2}+{\left|Q^{\alpha \beta }\right|}^{2}+P_{1}^{\alpha \beta },$$
where $Q^{\alpha \beta }$ is another function of *E* defined up to an arbitrary phase factor:(21)$$Q^{\alpha \beta }=\det\left({\hat{A}}^{\alpha \beta }-i{\hat{\Gamma }}^{\alpha }+i{\hat{\Gamma }}^{\beta }\right)$$
and $P_{1}^{\alpha \beta }$ is the following extra term, which is non-zero due to the non-Hermiticity of the matrix ${\hat{A}}^{\alpha \beta }$:(22)P1αβ=4detA^αβ2−Imuα†A^αβ−1uα1+iuβ†A^αβ−1uβ2+Reuα†A^αβ−1−A^αβ†−1uβuβ†A^αβ−1uα+Imuβ†A^αβ−1uαuα†A^αβ−1uβuβ†A^αβ†−1uβ.

From Equation (Equation 22) one can see that for two-terminal structures, i.e., for Hermitian matrix ${\hat{A}}^{\alpha \beta }$, $P_{1}^{\alpha \beta }$ turns to zero and we exactly arrive to the calculations from [15]. Indeed, Hermiticity of ${\hat{A}}^{\alpha \beta }$ implies Hermiticity of its inverse ${\left({\hat{A}}^{\alpha \beta }\right)}^{-1}$, which provides turning to zero of the second term of Equation (Equation 22) due to the cancellation of ${\left({\hat{A}}^{\alpha \beta }\right)}^{-1}$ and $({{\hat{A}}^{\alpha \beta }{)}^{†}}^{-1}$. The first and the third terms in Equation (Equation 22) also vanish because $a^{†}{\left({\hat{A}}^{\alpha \beta }\right)}^{-1}a\in R$ and $a^{†}{\left({\hat{A}}^{\alpha \beta }\right)}^{-1}bb^{†}{\left({\hat{A}}^{\alpha \beta }\right)}^{-1}a={\left|a^{†}{\left({\hat{A}}^{\alpha \beta }\right)}^{-1}b\right|}^{2}\in R$ for any $a,b\in C^{N}$ in the case of Hermitian ${\hat{A}}^{\alpha \beta }$.

Quantity $Q^{\alpha \beta }$ from Equation (Equation 21) can be understood as a characteristic determinant of some auxiliary Hamiltonian ${\hat{H}}_{aux}$: $Q^{\alpha \beta }=\det\left(E\hat{I}-{\hat{H}}_{aux}\right)$, where
(23)$${\hat{H}}_{aux}={\hat{H}}_{0}+\sum_{\sigma =1}^{M}{\hat{\delta }}^{\sigma }-i\sum_{\begin{array}{c} \sigma =1,\\ \sigma \neq \alpha \end{array}}^{M}{\hat{\Gamma }}^{\sigma }+i{\hat{\Gamma }}^{\alpha }.$$

This auxiliary Hamiltonian differs from the effective one in Equation (Equation 7) only in the sign of ${\hat{\Gamma }}_{\alpha }$, which represents that the incoming electron flow goes from the α-th lead. Thus, the expression for the transmission coefficient between the α-th and β-th leads of an arbitrary multi-terminal quantum conductor can be written in the following form:(24)$$T_{\alpha \beta }=\frac{{\left|P_{0}^{\alpha \beta }\right|}^{2}}{{\left|P_{0}^{\alpha \beta }\right|}^{2}+P_{1}^{\alpha \beta }+{\left|Q^{\alpha \beta }\right|}^{2}},$$

The Equations (Equation 23) and (Equation 24) represent the main result of our paper.

In two terminal systems, matrix ${\hat{A}}^{\alpha \beta }$ is Hermitian [15] and hence we have $P_{1}^{\alpha \beta }=0$. Transmission in this case is governed only by $P=P_{0}^{\alpha \beta }$ and $Q=Q^{\alpha \beta }$ functions. Real roots of *P* define energies of zero transmission (antiresonances) and real roots of *Q*—energies of unity transmission (resonances). In a spatially symmetric two-terminal quantum conductor $H_{aux}$ becomes PT-symmetric and at EPs, where its PT-symmetry breaking takes place, resonances coalesce [15]. Quantity $P_{1}^{\alpha \beta }$ of the form of Equation (Equation 22) arises from the non-Hermiticity of ${\hat{A}}^{\alpha \beta }$ due to non-zero coupling to more than two leads. One can show in this case that $P_{1}^{\alpha \beta }\geq 0$ (see Appendix A for details), which guarantees that $0\leq T_{\alpha \beta }\leq 1$ (for real energies).

### 3.2. Conditions for Perfect and Zero Transmission

According to Equation (Equation 24), real roots of $P_{0}^{\alpha \beta }$ determine energies of zero transmission $T_{\alpha \beta }$ as in the two-terminal case. Using Equation (Equation 19) one can get the following conditions for DQI to take place ($T_{\alpha \beta }=0$):
(25a)$$u_{\alpha }^{†}{\hat{B}}_{1}u_{\beta }=0,$$
(25b)$$u_{\alpha }^{†}{\hat{B}}_{2}u_{\beta }=0.$$

Here ${\hat{B}}_{1,2}$ are defined as Hermitian and anti-Hermitian parts of ${\left({\hat{A}}^{\alpha \beta }\right)}^{-1}$ respectively (see Equation (Equation 54) in Appendix A).

Perfect (unity-valued) resonances of $T_{\alpha \beta }$ are located at energies, which provide both $P_{1}^{\alpha \beta }=0$ and $Q^{\alpha \beta }=0$. Analyzing Equations (Equation 21) and (Equation 22) one can conclude that $T_{\alpha \beta }=1$ takes place if the following conditions are fulfilled simultaneously (see Appendix B for details):
(26a)$$u_{\alpha }^{†}{\hat{B}}_{2}u_{\alpha }=0,$$
(26b)$$u_{\beta }^{†}{\hat{B}}_{2}u_{\beta }=0,$$
(26c)$$u_{\alpha }^{†}{\hat{B}}_{1}u_{\alpha }=u_{\beta }^{†}{\hat{B}}_{1}u_{\beta },$$
(26d)$$\det{\hat{A}}^{\alpha \beta }\left[1+{\left(u_{\alpha }^{†}{\hat{B}}_{1}u_{\alpha }\right)}^{2}-{\left|u_{\beta }^{†}{\hat{B}}_{1}u_{\alpha }\right|}^{2}\right]=0.$$

These conditions can be easily interpreted. Indeed, matrix ${\hat{B}}_{2}$ is responsible for the coupling with all the rest leads except the α-th and β-th and hence the first two conditions ([Disp-formula FD26a-entropy-22-00459]) and ([Disp-formula FD26b-entropy-22-00459]) reflect effective decoupling from all that leads. Equation ([Disp-formula FD26c-entropy-22-00459]) requires symmetric coupling to the α-th and β-th lead and Equation ([Disp-formula FD26d-entropy-22-00459]) defines the resonant energy. It is important to check that conditions (26) do not lead to $P_{0}^{\alpha \beta }=0$, i.e., $u_{\alpha }^{†}{\hat{B}}_{1}u_{\beta }\neq 0$. Otherwise, we would have $P_{1}^{\alpha \beta }=Q^{\alpha \beta }=P_{0}^{\alpha \beta }=0$, which means the presence of a real eigenvalue of the effective Hamiltonian (i.e., real *S*-matrix pole), indicating the formation of a bound state in the continuum (BIC) [17]. Transmission coefficient at BIC, in general, is indeterminate and it can be derived only from the analysis of multiplicity of the roots of $P_{1}^{\alpha \beta }$, $Q^{\alpha \beta }$, and $P_{0}^{\alpha \beta }$ [15].

For illustration consider an example of a simple two-terminal (M=2) resonant tunnelling conductor with single state (N=1) of energy $\varepsilon_{0}$. In this case 1×1 matrix ${\hat{A}}^{12}=E-{\tilde{\varepsilon }}_{0}$ is Hermitian and hence ${\hat{B}}_{1}={\left({\hat{A}}^{12}\right)}^{-1}={\left(E-{\tilde{\varepsilon }}_{0}\right)}^{-1}$ and ${\hat{B}}_{2}=0$. Here ${\tilde{\varepsilon }}_{0}=\varepsilon_{0}+\delta^{1}+\delta^{2}$ is the hybridized eigenenrgy of the state. Therefore, conditions ([Disp-formula FD26a-entropy-22-00459]) and ([Disp-formula FD26b-entropy-22-00459]) are fulfilled identically. Condition ([Disp-formula FD26c-entropy-22-00459]) requires the equivalent coupling to the leads: $\gamma_{1}=\pm \gamma_{2}$ and condition ([Disp-formula FD26d-entropy-22-00459]) requires incident electron energy *E* to be equal to ${\tilde{\varepsilon }}_{0}$.

## 4. Three-Terminal Quantum Conductors: Illustrative Examples

In this section, we will apply the above-proposed formalism to study in detail different three-terminal systems and the change of their transport properties with the insertion of the third electrode. In this section we will work within the wide-band limit (WBL) [28] and use notation $\gamma_{i}^{\alpha }$ instead of $\sqrt{\pi \rho_{\alpha }}\gamma_{i}^{\alpha }$ as elements of coupling vectors $u_{\alpha }$ (see general Equation (Equation 13)) for simplicity.

### 4.1. Suppression of Transmission by the Third Electrode

It is well-known that coupling to electrodes in multi-terminal systems results in suppression of resonant tunneling in coherent transport [1], which arises from the imaginary part of the electrode self-energy. In the case of a three-terminal quantum conductor (inset in Figure 1a) consisting of a single state with energy $\varepsilon_{0}$ (resonant-tunneling transistor) all matrix and vector quantities, which appeared in general equations in the previous section, has dimension one, i.e., are just numbers. In the wide-band limit (WBL) [28], when we neglect the energy dependence of lead to conductor couplings $\gamma_{i}^{\alpha }=\gamma_{\alpha }$, one can easily derive expressions for $T_{12}$ and $T_{13}$ transmissions:(27)$$T_{12}\left(E\right)=\frac{4\gamma_{1}^{2}\gamma_{2}^{2}}{{\left(E-\varepsilon_{0}\right)}^{2}+{\left(\gamma_{1}^{2}+\gamma_{2}^{2}+\gamma_{3}^{2}\right)}^{2}},\;T_{13}\left(E\right)=\frac{4\gamma_{1}^{2}\gamma_{3}^{2}}{{\left(E-\varepsilon_{0}\right)}^{2}+{\left(\gamma_{1}^{2}+\gamma_{2}^{2}+\gamma_{3}^{2}\right)}^{2}}.$$

From these equations one can see that $\gamma_{3}$ can be interpreted as an additional dephasing/dissipation, which suppresses the lead 1 to lead 2 tunneling. Coupling $\gamma_{2}$ acts similarly for the lead 1 to lead 3 tunneling process. Clearly, $\gamma_{2}/\gamma_{3}$ ratio defines the ratio of transmission coefficients $T_{12}/T_{13}$. Figure 1 illustrates Equation (Equation 27) for different parameters.

In the case of a two-state quantum conductor, transmission behavior becomes substantially more complicated. Consider a two-site model with the following Hamiltonian in an atomic orbital basis
(28)$${\hat{H}}_{0}=\left(\begin{array}{cc} \varepsilon_{0} & \tau \\ \tau & \varepsilon_{0} \end{array}\right),$$
which is connected equally to two leads (Figure 2a):(29)$$u_{1}={\left(\gamma \;0\right)}^{⊤},\;u_{2}={\left(0\;\gamma \right)}^{⊤}.$$

Without the third electrode this system has
(30)$$P_{0}^{12}=2\gamma^{2}\tau ,\;Q^{12}={\left(E-\varepsilon_{0}\right)}^{2}-\tau^{2}+\gamma^{4},$$
and surely $P_{1}^{12}\equiv 0$. The auxiliary Hamiltonian in this two-terminal configuration is PT-symmetric:(31)$${\hat{H}}_{aux}=\left(\begin{array}{cc} \varepsilon_{0}-i\gamma^{2} & \tau \\ \tau & \varepsilon_{0}+i\gamma^{2} \end{array}\right),$$
and it can possess an EP at $\gamma^{2}=\tau$, which corresponds to the resonance coalescence phenomenon [14].

Insertion of the third electrode with coupling vector
(32)$$u_{3}={\left(\gamma_{1}\;\gamma_{2}\right)}^{⊤},$$
gives the transmission coefficient $T_{12}$ of the form Equation (Equation 24) with
(33)$$\begin{array}{cc} P_{0}^{12} & =2\gamma^{2}\left(\tau -i\gamma_{1}\gamma_{2}\right),\\ P_{1}^{12} & =4\gamma^{2}\left\{\left[{\left(E-\varepsilon_{0}\right)}^{2}+\gamma^{4}\right]\gamma_{1}^{2}+2\left(E-\varepsilon_{0}\right)\gamma_{1}\gamma_{2}\tau +\gamma_{2}^{2}\tau^{2}\right\},\\ Q^{12} & ={\left(E-\varepsilon_{0}\right)}^{2}+\gamma^{4}+\gamma^{2}\left(\gamma_{2}^{2}-\gamma_{1}^{2}\right)+i\left(E-\varepsilon_{0}\right)\left(\gamma_{1}^{2}+\gamma_{2}^{2}\right)+\left(2i\gamma_{1}\gamma_{2}-\tau \right)\tau . \end{array}$$

From Equation (Equation 33), one can see that $P_{0}^{12}$ is a non-zero constant either with or without the third electrode and hence DQI is not supposed to take place in this system. Insertion of the third electrode, however, makes roots of $Q^{12}$ to be complex, which results in suppression of perfect transmission resonances. This can be understood as non-spontaneous PT-symmetry breaking of the underlying auxiliary Hamiltonian induced by the external influence of the third lead. Additionally, we have $P_{1}^{12}\neq 0$ for any real energy, which also decreases resonant transmission maxima. Figure 2b shows energy dependence of $|Q^{12}|$ in two- and three-terminal configurations and $P_{1}^{12}$ in three-terminal configuration. One can see that with the third electrode insertion $Q^{12}$ and $P_{1}^{12}$ become strictly non-zero at real energies.

### 4.2. Quantum Interference Transistor

Resonance coalescence effect and DQI formation were proposed to be an efficient mechanism for current switching in a quantum interference transistor [29]. It was shown there that these phenomena take place in a system of two degenerate states of opposite parity (with respect to the mirror symmetry reflecting source and drain electrodes to each other). Gate was assumed to have only an electrostatic influence on the system, which resulted in the lifting of degeneracy. However, non-zero coupling to the third (gate) electrode is almost inevitable in a real system, and, as we have shown above, this leads to the degradation of interference features in source-to-drain quantum transport. Hence, the question arises—is it possible to find the configuration of the gate electrode coupling, which would have minimal impact on the interference transport?

Consider a two-state system with two degenerate states of different parity, which can be lifted by the gate electrode potential (Figure 3a). Its Hamiltonian in a molecular orbital basis can be written as
(34)$${\hat{H}}_{0}=\left(\begin{array}{cc} \varepsilon_{0}-\frac{\Delta }{2} & 0\\ 0 & \varepsilon_{0}+\frac{\Delta }{2} \end{array}\right).$$

Here $\varepsilon_{0}$ is the energy of degenerate states, and Δ is the energy split induced by the gate. Different parity of these states manifests itself in the coupling vectors to the source and drain electrodes (assume the first and the second electrodes for definiteness):(35)$$u_{1}={\left(\gamma \;\gamma \right)}^{⊤},\;u_{2}={\left(\gamma \;-\gamma \right)}^{⊤}.$$

Without coupling to the gate electrode taken into account, we have for this system:(36)$$P_{0}^{12}=2\gamma^{2}\Delta ,\;Q^{12}={\left(E-\varepsilon_{0}\right)}^{2}-\frac{\Delta^{2}}{4}+4\gamma^{4},$$
and $P_{1}^{12}\equiv 0$. The auxiliary Hamiltonian is PT-symmetric [29]:(37)$${\hat{H}}_{aux}=\left(\begin{array}{cc} \varepsilon_{0}-\frac{\Delta }{2} & 2i\gamma^{2}\\ 2i\gamma^{2} & \varepsilon_{0}+\frac{\Delta }{2} \end{array}\right),$$
and its EP (i.e., resonance coalescence) takes place at $\Delta =4\gamma^{2}$. The key feature of this system is that its transmission turns identically into zero in the case of degenerate states as $P_{0}^{12}\equiv 0$ for Δ=0. This provides, for instance, theoretically unbounded logarithmic transconductance and $I_{on}/I_{off}$ ratio [29].

Taking into account non-zero coupling to the gate (third) electrode with
(38)$$u_{3}={\left(\gamma_{1}\;\gamma_{2}\right)}^{⊤},$$
provides
(39)$$\begin{array}{cc} P_{0}^{12} & =2\gamma^{2}\left[\Delta -i\left(\gamma_{1}^{2}-\gamma_{2}^{2}\right)\right],\\ P_{1}^{12} & =\gamma^{2}\left\{\Delta^{2}{\left(\gamma_{1}-\gamma_{2}\right)}^{2}+4\Delta \left(E-\varepsilon_{0}\right)\left(\gamma_{1}^{2}-\gamma_{2}^{2}\right)+4\left[{\left(E-\varepsilon_{0}\right)}^{2}+4\gamma^{4}\right]{\left(\gamma_{1}+\gamma_{2}\right)}^{2}\right\},\\ Q^{12} & ={\left(E-\varepsilon_{0}\right)}^{2}-\frac{\Delta^{2}}{4}+4\gamma^{4}-4\gamma^{2}\gamma_{1}\gamma_{2}+\frac{i}{2}\Delta \left(\gamma_{1}^{2}-\gamma_{2}^{2}\right)+i\left(E-\varepsilon_{0}\right)\left(\gamma_{1}^{2}+\gamma_{2}^{2}\right). \end{array}$$

Similar to Equation (Equation 33), here we also see that the third electrode prevents $Q^{12}$ and $P_{1}^{12}$ from turning to zero at real energies, which results in suppression of the resonant tunneling. On the other hand, from Equation (Equation 39), one can see that transmission $T_{12}$ can still turn to zero identically even with non-zero coupling to the third electrode if $\gamma_{1}=\pm \gamma_{2}$.

The presence of tunneling coupling with the gate electrode allows parasitic leakage currents, which arise from non-zero $T_{13}$ and $T_{23}$ transmission coefficients. In the optimal case, for $\gamma_{1}=\gamma_{2}$ one can derive that
(40)$$T_{13}=4\frac{\gamma_{1}^{2}}{\gamma^{2}}\times \frac{{\left(E-\varepsilon_{0}\right)}^{2}+4\gamma^{4}}{\Delta^{2}}\times T_{12},\;T_{23}=\frac{\gamma_{1}^{2}}{\gamma^{2}}\times T_{12}.$$

The key difference between $T_{13}$ and $T_{23}$ arises from the fact that for $\gamma_{1}=\gamma_{2}$, the third (gate) electrode is attached in the same configuration as the first (source) one. Therefore, transmission $T_{23}$ resembles the transmission $T_{12}$ as states are coupled with different parity to the third and second electrodes (as in the case of the first and second electrodes either). On the other hand, coupling to the first and the third electrodes have the same parity, and hence transmission $T_{13}$ differs dramatically from the $T_{12}$. In the case of $\gamma_{1}=-\gamma_{2}$ one should swap expressions for $T_{13}$ and $T_{23}$, obviously.

From Equation (Equation 40) one can see that $T_{23}$ scales as the square of $\gamma_{1}/\gamma$ ratio and hence blocking this leakage essentially requires $\gamma_{1}\ll \gamma$. Transmission $T_{13}$ has an additional factor, which “blows up” at Δ→0. At first sight, it makes source-gate leakage dominant in the “off” state of the transistor. However, if Δ=0 corresponds to low potential on the gate electrode $V_{3}\approx 0$ (see Figure 3a), then leakage current between source and gate will be negligible regardless of nonzero transmission coefficient because of zero voltage bias between these electrodes. In the case of $\gamma_{1}=-\gamma_{2}$, the same argument is applicable, but the source and the drain electrodes must be swapped. Thus, one can conclude that tunneling coupling to the gate electrode has the least impact on the interference transport in the quantum interference transistor, proposed in [29], if the configuration of this coupling is the same as for the source electrode but with a much smaller amplitude. Figure 3 illustrates evolution of the transmission coefficients with varying Δ for $\gamma_{1}=\gamma_{2}=0.5\gamma$.

## 5. Three-Terminal Quantum Conductors: Comparison With Two-Terminal Configuration

### 5.1. Perfect Transmission

In the case of three-terminal system (M=3), one can explicitly calculate the inverse of non-Hermitian matrix ${\hat{A}}^{\alpha \beta }$ from Equation (Equation 14) in terms of coupling vector $u_{3}$ and its Hermitian part ${\hat{A}}_{1}=E\hat{I}-{\hat{H}}_{0}-\sum_{\sigma =1}^{3}{\hat{\delta }}^{\sigma }$, which also corresponds to the two-terminal (α and β) configuration except for ${\hat{\delta }}^{3}$ term. This term reflects the hybridization of the system by the third electrode and can be neglected in WBL approximation. Without loss of generality we assume α=1 and β=2 and get:(41)$${\left({\hat{A}}^{12}\right)}^{-1}={\hat{A}}_{1}^{-1}-\frac{u_{3}^{†}{\hat{A}}_{1}^{-1}u_{3}}{1+{\left(u_{3}^{†}{\hat{A}}_{1}^{-1}u_{3}\right)}^{2}}{\hat{A}}_{1}^{-1}u_{3}u_{3}^{†}{\hat{A}}_{1}^{-1}-\frac{i}{1+{\left(u_{3}^{†}{\hat{A}}_{1}^{-1}u_{3}\right)}^{2}}{\hat{A}}_{1}^{-1}u_{3}u_{3}^{†}{\hat{A}}_{1}^{-1}.$$

From Equation (Equation 41) one can get exact expressions for Hermitian and anti-Hermitian parts of ${\left({\hat{A}}^{12}\right)}^{-1}$ (i.e., for ${\hat{B}}_{1,2}$ correspondingly) and use them to analyze conditions for perfect transmission in Equations (26). In the case of three-terminal systems conditions in Equations ([Disp-formula FD26a-entropy-22-00459]) and ([Disp-formula FD26b-entropy-22-00459]) reduce to
(42a)$$u_{1}^{†}{\hat{A}}_{1}^{-1}u_{3}=0,$$
(42b)$$u_{2}^{†}{\hat{A}}_{1}^{-1}u_{3}=0.$$

Using Equations ([Disp-formula FD42a-entropy-22-00459]) and ([Disp-formula FD42b-entropy-22-00459]) one can see that conditions in Equations ([Disp-formula FD26c-entropy-22-00459]) and ([Disp-formula FD26d-entropy-22-00459]) can be rewritten with ${\hat{A}}_{1}^{-1}$ instead of ${\hat{B}}_{1}$, i.e., they become the same as in the case of two leads except for taking into account hybridization from the third electrode (${\hat{\delta }}^{3}$). Thus, within WBL approximation we get that three-terminal quantum conductor has perfect (unity-valued) transmission resonances of $T_{12}$ transmission for the same energies as in the two terminal case if conditions in Equations ([Disp-formula FD42a-entropy-22-00459]) and ([Disp-formula FD42b-entropy-22-00459]) are fulfilled. It should be noted, for clarity, that perfect transmission $T_{12}$ implies zero transmission $T_{13}$ because of particle flow conservation. One can check this directly using Equations ([Disp-formula FD42a-entropy-22-00459]) and ([Disp-formula FD42b-entropy-22-00459]). Transmission $T_{13}$ turns to zero, if $P_{0}^{13}$ does so. The latter can be written as
(43)$$P_{0}^{13}=2\det{\hat{A}}_{1}\left(u_{3}^{†}{\hat{A}}_{1}^{-1}u_{1}+iu_{2}^{†}{\hat{A}}_{1}^{-1}u_{2}u_{3}^{†}{\hat{A}}_{1}^{-1}u_{1}-iu_{3}^{†}{\hat{A}}_{1}^{-1}u_{2}u_{2}^{†}{\hat{A}}_{1}^{-1}u_{1}\right),$$
and it is clear that conditions in Equations ([Disp-formula FD42a-entropy-22-00459]) and ([Disp-formula FD42b-entropy-22-00459]) imply $P_{0}^{13}=0$.

Using conditions in Equations ([Disp-formula FD42a-entropy-22-00459]) and ([Disp-formula FD42b-entropy-22-00459]) instead of analyzing full expressions for $P_{1}^{\alpha \beta }$ and $Q^{\alpha \beta }$ is typically a much easier task as will be illustrated by the following examples. Single-state quantum conductor surely cannot possess perfect transmission in the presence of the third electrode (see Equation (Equation 27)). Conditions ([Disp-formula FD42a-entropy-22-00459]) and ([Disp-formula FD42b-entropy-22-00459]) cannot be satisfied in this case as their left-hand side is non-zero constant. Then, consider a two-site system with the bare Hamiltonian in Equation (Equation 28), which is coupled to three electrodes by (see Figure 4a)
(44)$$u_{1}={\left(\gamma_{1}^{\left(1\right)}\;\gamma_{2}^{\left(1\right)}\right)}^{⊤},\;u_{2}={\left(\gamma_{1}^{\left(2\right)}\;\gamma_{2}^{\left(2\right)}\right)}^{⊤},\;u_{3}={\left(\gamma_{1}^{\left(3\right)}\;\gamma_{2}^{\left(3\right)}\right)}^{⊤}.$$

Conditions in Equations ([Disp-formula FD42a-entropy-22-00459]) and ([Disp-formula FD42b-entropy-22-00459]) in this case can be written as
(45)$$\begin{array}{c} \gamma_{1}^{\left(3\right)}\left[\left(E-\varepsilon_{0}\right)\gamma_{1}^{\left(1\right)}+\tau \gamma_{2}^{\left(1\right)}\right]+\gamma_{2}^{\left(3\right)}\left[\left(E-\varepsilon_{0}\right)\gamma_{2}^{\left(1\right)}+\tau \gamma_{1}^{\left(1\right)}\right]=0,\\ \gamma_{1}^{\left(3\right)}\left[\left(E-\varepsilon_{0}\right)\gamma_{1}^{\left(2\right)}+\tau \gamma_{2}^{\left(2\right)}\right]+\gamma_{2}^{\left(3\right)}\left[\left(E-\varepsilon_{0}\right)\gamma_{2}^{\left(2\right)}+\tau \gamma_{1}^{\left(2\right)}\right]=0. \end{array}$$

These conditions can be considered as a homogeneous system of linear equations with respect to $\gamma_{1}^{\left(3\right)}$ and $\gamma_{2}^{\left(3\right)}$. This system has non-trivial solutions if E=±τ or $\gamma_{1}^{\left(1\right)}\gamma_{2}^{\left(2\right)}=\gamma_{1}^{\left(2\right)}\gamma_{2}^{\left(1\right)}$. Solution of Equation (Equation 45) in this case must satisfy $\gamma_{1}^{\left(3\right)}=\mp \gamma_{2}^{\left(3\right)}$. Under these restrictions one can analyze conditions in Equations ([Disp-formula FD26c-entropy-22-00459]) and ([Disp-formula FD26d-entropy-22-00459]) and get the full set of conditions, which must be fulfilled simultaneously to get a perfect transmission resonance in $T_{12}$:(46)$$E=\varepsilon_{0}\pm \tau ,\;\gamma_{1}^{\left(1\right)}\gamma_{2}^{\left(2\right)}=\gamma_{1}^{\left(2\right)}\gamma_{2}^{\left(1\right)},\;\left|\gamma_{1}^{\left(1\right)}\pm \gamma_{2}^{\left(1\right)}\right|=\left|\gamma_{1}^{\left(2\right)}\pm \gamma_{2}^{\left(2\right)}\right|,\;\gamma_{1}^{\left(3\right)}=\mp \gamma_{2}^{\left(3\right)}.$$

Figure 4b shows and example of the transmission coefficient of the system with particular parameters, which satisfy Equations (Equation 46).

Consider now another example – a linear three-site quantum conductor (Figure 5a). It has the following bare Hamiltonian:(47)$${\hat{H}}_{0}=\left(\begin{array}{ccc} \varepsilon_{0} & \tau & 0\\ \tau & \varepsilon_{0} & \tau \\ 0 & \tau & \varepsilon_{0} \end{array}\right).$$

If two leads are attached in a linear configuration (as shown in Figure 5a for the lead 1 and 2) with
(48)$$u_{1}={\left(\gamma \;0\;0\right)}^{⊤},\;u_{2}={\left(0\;0\;\gamma \right)}^{⊤},$$
then perfect transmission will take place at three resonant energies: $E=\varepsilon_{0}\pm \sqrt{2\tau^{2}-\gamma^{4}}$ and $E=\varepsilon_{0}$ if $\gamma^{2}<\sqrt{2}\tau$ [14]. These unity transmission resonances coalesce at $\gamma^{2}=\sqrt{2}$, which corresponds to an EP of the underlying auxiliary Hamiltonian. Here we again use $\gamma_{i}^{\alpha }$ instead of $\sqrt{\pi \rho_{\alpha }}\gamma_{i}^{\alpha }$ as elements of vectors $u_{\alpha }$ and treat these couplings as energy-independent constants within WBL approximation.

Insertion of the third lead with the coupling vector
(49)$$u_{3}={\left(\gamma_{1}\;\gamma_{2}\;\gamma_{3}\right)}^{⊤}$$
results, in general, in suppression of the tunneling resonances between the first and the second leads. However, one can utilize Equations ([Disp-formula FD42a-entropy-22-00459]) and ([Disp-formula FD42b-entropy-22-00459]) to figure out what particular coupling configuration in Equation (Equation 49) will allow perfect transmission resonances in $T_{12}$ with the third electrode connected to the system. It turns out that perfect transmission takes place at $E=\varepsilon_{0}\pm \sqrt{2\tau^{2}-\gamma^{4}}$ for $\gamma_{1}=\gamma_{3}=\mp \gamma_{2}\tau /\sqrt{2\tau^{2}-\gamma^{4}}$ correspondingly and at $E=\varepsilon_{0}$ – for $\gamma_{2}=0$ and $\gamma_{1}=\gamma_{3}$. Detailed analysis of these equations is presented in Appendix C. Figure 5 shows plots of $T_{12}$ in two- and three-terminal configurations having a perfect resonance. Surely, $T_{13}=0$ at perfect resonance of $T_{12}$. Parameters are chosen as follows: $\gamma^{2}=\frac{1}{2}\tau$ in all cases, $\gamma_{2}^{2}=\tau$ in cases (b) and (d), and $\gamma_{1}^{2}=\tau$ in case (c).

### 5.2. Transmission and Bound States in the Continuum

There is another interesting phenomenon, which takes place in three-terminal configurations within WBL approximation. The system with a BIC in a two-terminal configuration does not change its transmission coefficient at BIC if the third electrode is attached. BIC is such a localized state of the system, which energy lies within the spectrum of continuous states and, for some reason, has zero matrix elements with them [17]. Suppose that some *i*-th eigenstate of the system with energy $\varepsilon_{i}$ is effectively decoupled from the first and second electrodes, but has a non-zero coupling to the third one: $u_{1,i}=u_{2,i}=0$ and $u_{3,i}\neq 0$. In this case, the following scalar products can be treated as energy-independent constants in the vicinity of $E=\varepsilon_{i}$:(50)$$u_{\sigma }^{†}{\hat{A}}_{1}^{-1}u_{\sigma^{\prime }}=\sum_{\begin{array}{c} j=1,\\ j\neq i \end{array}}^{N}\frac{u_{\sigma ,j}^{*}u_{\sigma^{\prime },j}}{E-\varepsilon_{j}}\approx c_{\sigma \sigma^{\prime }}=const,$$
where σ=1,2 or $\sigma^{\prime }=1,2$. Utilizing Equations (Equation 41) and (Equation 50) we can deduce that in the vicinity of $E=\varepsilon_{i}$ scalar products $u_{\sigma }^{†}{\hat{B}}_{2}u_{\sigma^{\prime }}$ for σ=1,2 or $\sigma^{\prime }=1,2$ are linear in $E-\varepsilon_{i}$ and hence turn to zero exactly at BIC energy. Moreover, one can show that $u_{\sigma }^{†}{\hat{B}}_{1}u_{\sigma^{\prime }}\approx u_{\sigma }^{†}{\hat{A}}_{1}^{-1}u_{\sigma^{\prime }}+O\left(E-\varepsilon_{i}\right)$ for σ=1,2 and $\sigma^{\prime }=1,2$. Exactly at $E=\varepsilon_{i}$ these products have the same values as in the two-terminal configuration (for $u_{3}=0$). Thus, from Equations (Equation 19)–(Equation 22) we see that the transmission coefficient exactly at BIC energy does not change if the third electrode is inserted.

As an illustrative example for this observation, we consider a three-site model (Figure 6a), which is known to possess BICs with zero, unity, or intermediate value of the transmission coefficient depending on particular system parameters values [15]. The Hamiltonian of the isolated model system is the following:(51)$${\hat{H}}_{0}=\left(\begin{array}{ccc} \varepsilon_{0} & \tau_{a} & \tau_{b}\\ \tau_{a} & \varepsilon & \eta \\ \tau_{b} & \eta & \varepsilon \end{array}\right).$$

Without loss of generality, we can assume $\varepsilon_{0}=0$, i.e., choose it as energy origin.

Coupling vectors $u_{1,2}$ we assume to be of the following form:(52)$$u_{1}=u_{2}={\left(\gamma \;0\;0\right)}^{⊤}.$$

In a two-terminal configuration ($u_{3}=0$) this system has BICs with different transmission values [15]. For instance, consider the following particular cases:1.$\tau_{b}=\frac{1}{2}\tau_{a}$, $\varepsilon =\tau_{a}$, and η=0 give BIC at $E=E_{BIC}=\varepsilon =\tau_{a}$ and transmission $T_{12}\left(E_{BIC}\right)=0$,2.$\tau_{b}=\varepsilon =\eta =\tau_{a}$ give BIC at $E=E_{BIC}=\varepsilon -\eta =0$ and transmission $0<T_{12}\left(E_{BIC}\right)<1$,3.$\tau_{b}=\eta =\tau_{a}$ and $\varepsilon =\varepsilon_{0}=0$ give BIC at $E=E_{BIC}=\varepsilon -\eta =-\tau_{a}$ and transmission $T_{12}\left(E_{BIC}\right)=1$.

Detailed discussion of these BICs is presented in Appendix D. Figure 6b–d show corresponding two-terminal transmission coefficient for $\gamma^{2}=\frac{1}{2}\tau_{a}$. Insertion of the third lead with non-zero coupling $u_{3}={\left(\gamma_{1}\;\gamma_{2}\;\gamma_{3}\right)}^{⊤}$ surely modifies the transmission except for $E=E_{BIC}$. Exact expressions for $P_{0}^{12}$, $P_{1}^{12}$, and $Q^{12}$ for this structure are shown in Appendix D. Blue solid lines in Figure 6b–d show $T_{12}$ transmission spectrum for $\gamma_{1}^{2}=\tau_{a}$, $\gamma_{2}^{2}=4\tau_{a}$, and $\gamma_{3}^{2}=9\tau_{a}$ respectively. Exactly at $E=E_{BIC}$ transmission does not change with the insertion of the third electrode.

## 6. Summary and Discussion

In this paper, we have presented a novel approach to a multi-terminal quantum transport description using non-Hermitian Hamiltonians. A traditional treatment based on scattering matrix formalism [21,22] made it possible to establish general symmetry relations for conductance and elucidated the difference between current and voltage probes, which, e.g., can result in formally negative values of the measured resistance. The problem, which we have addressed in the present paper and that stands beyond multi-terminal scattering matrix theory [21,22] is that terminal-to-terminal transparencies depend not only on electrode location but also on the peculiarities of quantum conductor couplings with electrodes. In particular, a detailed investigation of the molecular conductance dependence on the location of additional electrode could be of high interest for future experimental studies.

Our method generalizes the result of [15] and provides exact rules of finding perfect (even in the presence of extra electrodes) and zero transmission points, which essentially supplements the results of ab initio modeling of multi-terminal molecular devices (e.g., [18]). In the case of three-terminal systems, these rules can be simplified dramatically and provide an illustrative correspondence with two-terminal systems. It should be noted that we have used the tight-binding approach without taking electron-electron interactions into account. Coulomb interactions depend on the electron density, and hence it should be low throughout the quantum conductor to make our theory valid. This implies that couplings to the leads must be sufficiently high to prevent charge accumulation and, e.g., the Coulomb blockade effect. On the other hand, the tight-binding approach (and WBL approximation in particular) requires that tunneling matrix elements between the leads and the quantum conductor should be less than hopping integrals inside the isolated electrode. These two restrictions on the coupling strength define the domain of applicability of our method. The fact that our theory can be implemented to single-molecular conductors is proved by the well-established (by comparison with experiments and ab initio calculations) applicability (at least qualitatively) of a simple Hückel molecular orbital method [30,31,32].

Presented results may be of interest for the development of designing principles of active molecular electronic devices (e.g., transistors), which are under active experimental [33,34,35,36] and ab initio modeling [37,38,39] study today. Among these investigations, the most relevant to our theory are those, which consider molecules with complicated coupling to the leads [35,36,39]. In particular, the theory developed in the present paper gives a big deal for making design rules of molecular electronic devices based on quantum interference effects, such as, for instance, [29,40]. The transmission coefficient based approach for quantum transport treatment is also suitable for the description of thermoelectric properties of quantum conductors [41,42]. Interference effects in this context are important as they can provide a great enhancement in thermoelectric effect [43].

In Ref. [29] we have proposed a new molecular quantum interference transistor (MQIT) with extremely high logarithmic transconductance and high “on”/“off” current ratio, which operates near EP of an OQS comprised of a molecule and electrodes. Control of this transistor is realized by a capacitively coupled gate, which is electrically decoupled from the molecule, similarly to a gate in an ordinary metal-oxide-semiconductor field-effect transistor (MOSFET). A perfect silicon oxide layer provides electrical isolation in the latter case. Such a dielectric layer is not available for electronics based on III-V semiconductor heterostructures, where Schottky barrier field-effect transistors (SBFET) are used. However, unavoidable gate leakage currents in SBFETs are not critical at high frequencies, at which MOSFET circuits possess significant power consumption. In molecular electronics formation of a perfect electrically isolated gate in MQIT will essentially complicate technological flow. Hence gate leakage currents should be taken into account as well. Here we have applied newly developed formalism to study simple two-level MQIT with electrically coupled gate and have shown that by a proper choice of molecule couplings with gate electrodes, leakage currents can be made insignificant for transistor operation. More complicated and realistic MQIT structures, as well as multi-terminal current-voltage characteristics at finite bias, will be studied elsewhere.

## Figures and Tables

**Figure 1 entropy-22-00459-f001:**
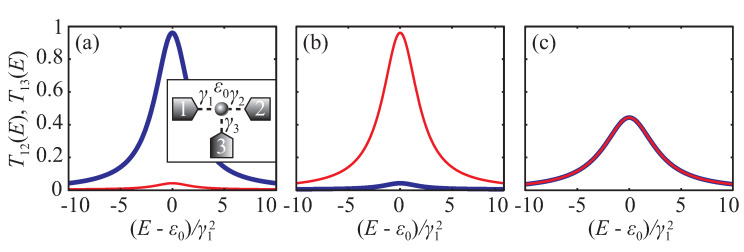
Transmission coefficients $T_{12}$ (blue thick line) and $T_{13}$ (red thin line) of a single-state quantum conductor for $\gamma_{2}=\gamma_{1}$ and $\gamma_{3}=0.2\gamma_{1}$ (**a**), $\gamma_{2}=0.2\gamma_{1}$ and $\gamma_{3}=\gamma_{1}$ (**b**) and $\gamma_{2}=\gamma_{3}=\gamma_{1}$ (**c**). Inset in plot (**a**): schematic view of the single-state quantum conductor connected to three electrodes.

**Figure 2 entropy-22-00459-f002:**
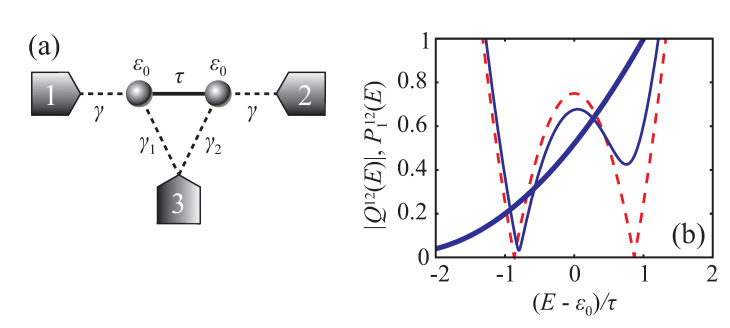
Schematic view of the two-site quantum conductor connected to three electrodes (**a**). Energy dependence of $|Q^{12}|$ in three-terminal (thin solid blue line) and two-terminal (dashed red line) configurations and $P_{1}^{12}$ in three-terminal configuration (thick solid blue line) (**b**). Parameters are the following: $\gamma =\sqrt{\tau /2}$, $\gamma_{1}=0.2\sqrt{\tau }$, and $\gamma_{2}=0.5\sqrt{\tau }$ (in two-terminal configurations we set $\gamma_{1}=\gamma_{2}=0$).

**Figure 3 entropy-22-00459-f003:**
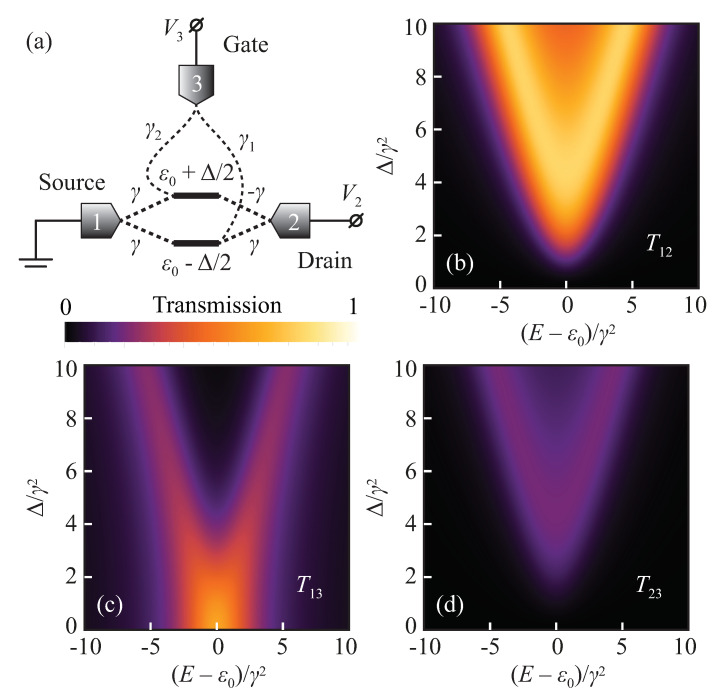
Schematic view of the quantum interference transistor based on a two-level system (**a**). Evolution of the transmission coefficients $T_{12}$ (**b**), $T_{13}$ (**c**) and $T_{23}$ (**d**) with varying energy splitting Δ. One can see that tunneling from source (lead 1) to drain (lead 2) is completely suppressed in degenerate regime (Δ=0). Parameters are chosen as following: $\gamma_{1}=\gamma_{2}=0.5\gamma$.

**Figure 4 entropy-22-00459-f004:**
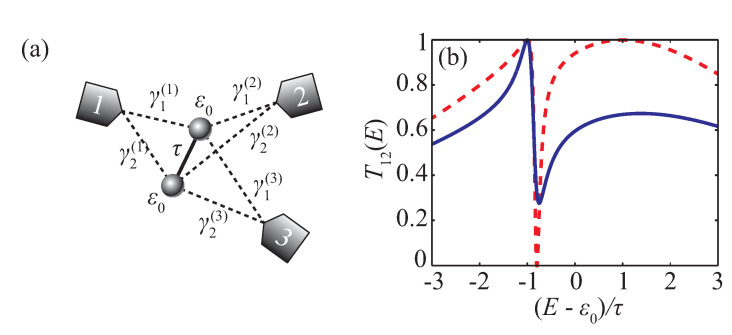
Schematic view of the two-site system (**a**). Transmission coefficient $T_{12}$ with $u_{3}\neq 0$ (blue solid line) and $u_{3}=0$ (red dashed line) in configurations, which provide perfect resonance at $E=\varepsilon_{0}-\tau$ (**b**). Parameters are $\gamma_{1}^{\left(1\right)}=-\gamma_{1}^{\left(2\right)}=\sqrt{\tau /2}$, $\gamma_{2}^{\left(1\right)}=-\gamma_{2}^{\left(2\right)}=\sqrt{2\tau }$, and $\gamma_{1}^{\left(3\right)}=\gamma_{2}^{\left(3\right)}=\sqrt{\tau /2}$.

**Figure 5 entropy-22-00459-f005:**
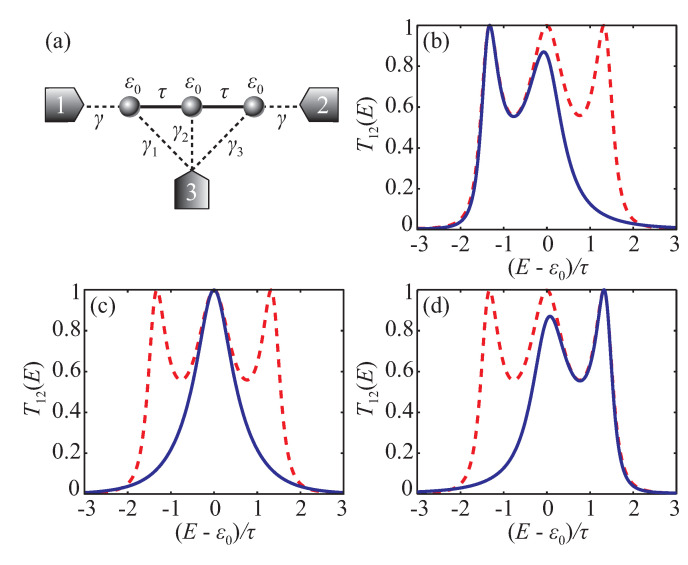
Schematic view of the linear three-site system (**a**). Transmission coefficient $T_{12}$ with $u_{3}\neq 0$ (blue solid line) and $u_{3}=0$ (red dashed line) in configurations, which provide perfect resonance at $E=\varepsilon_{0}-\sqrt{2\tau^{2}-\gamma^{4}}$ (**b**), $E=\varepsilon_{0}$ (**c**), and $E=\varepsilon_{0}+\sqrt{2\tau^{2}-\gamma^{4}}$ (**d**).

**Figure 6 entropy-22-00459-f006:**
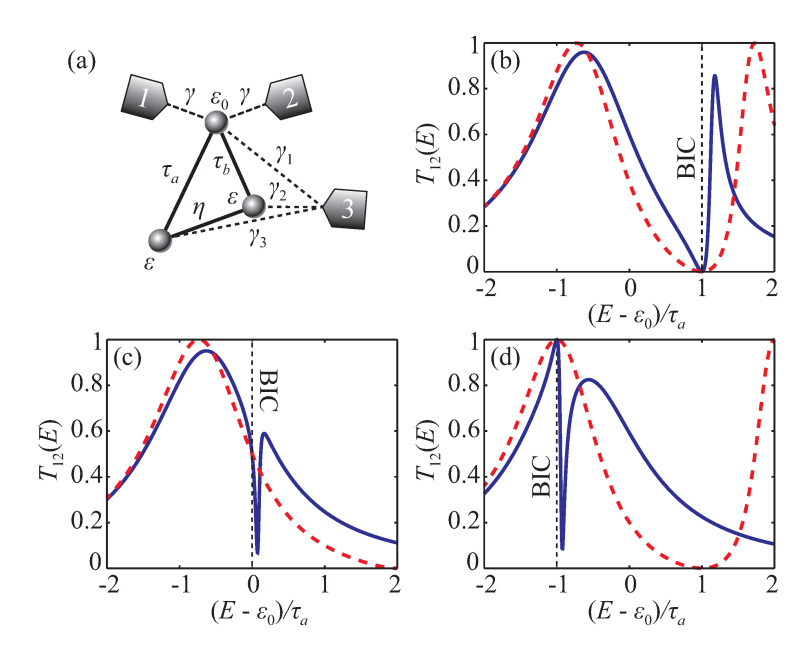
Schematic view of the three-site model system (**a**). Transmission coefficient $T_{12}$ with $u_{3}\neq 0$ (blue solid line) and $u_{3}=0$ (red dashed line) in configurations, which provide BIC with different transmission coefficient values (**b**–**d**). Vertical dashed lines indicate BIC energy.

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
