# Peer review of "Non-Hermitian Hamiltonians and Quantum Transport in Multi-Terminal Conductors"

_entropy, 2020, doi:10.3390/e22040459_

Round 1
Reviewer 1 Report
This paper describes theoretical calculations that examine transport properties of molecular tunnel junctions. The simulation particularly focuses on three-terminal junctions that are used for electrostatic gating of molecular orbital in junction. As an experimentalist,I believe the simulation should be linked with related experiments (the other way around, as well). In this regard, the authors should address the following comments for publication of this work.- Transmission function-based quantum transport is actively and heavily used in the field of molecular electronics and thermoelectrics. However, the work does not describe this (even not briefly). I strongly recommend the authors at least add a couple of sentences highlighting this and incorporate the following “recent” references: Chem. Rev. 2017, 117, 5, 4248-4286; J. Am. Chem. Soc. 2018, 140, 38, 12303-12307; Adv. Electronic Mater. 2020, 6, 2, 1901157; Nano Lett. 2018, 18, 12, 7715-7718; J. Mater. Chem. A, 2019,7, 14419-14446; J. Mater. Chem. C, 2016,4, 8842-8858; Nat. Nanotechnol. 8, 399–410(2013); ACS Cent. Sci. 2019, 5, 12, 1975-1982; Science, 2007, 315, 5818, 1568-1571; Chem. Mater. 2019, 31, 15, 5973-5980; Chem. Soc. Rev., 2016,45, 4285-4306.
- What type of experimental work in the literature can be most relevant to their calculation? If this discussion is added, this work would be of interest to not only theoreticians but also experimentalists.
- Similarly, it would be great if the authors can suggest future work for experimentalists to try based on their calculation.
Reviewer 2 Report
Non-Hermitian Hamiltonians and quantum transport in multi-terminal conductors by Shubin et al.
The manuscript computes the tunnelling transmission coefficient through open quantum cavities connected to multiple leads. This work generalises the authors’ earlier article from tunnelling transmission between 2 leads to a system with M leads. Some of the findings are made possible through the original adaptation of the non-Hermitian formalism to open quantum systems. The key finding of this study is that additional leads generally decrease the probability of tunnelling transmission and effectively act as a scattering source. The authors compute the resonances and anti-resonances pertaining to a number of illustrative examples.
The manuscript is generally well-written, the derivation is sound and should be published.
Major comments:
- The P_1^{\alpha\beta} term in Eq.22 is interpreted as a “dephasing term” due to contacts.
Dephasing generally implies an energy loss whereas, here, the authors presumably mean a delay along the transmission path. Inelastic scattering however is not part of the assumptions included in the calculation. The Phsyics would be clearer the physical meaning of the 3 terms in the P_1^{\alpha\beta} term (Eq.22) could be presented and discussed. Also, as written, I doubt it will be obvious to the reader why all 3 terms in the sum cancel when operator A is Hermitian.
Along the same line, the Landauer-Buttiker formula accounts for “non-phase destroying events” so it is incorrect to say, in the introduction of the paper, that this equation constitutes the “emergence of inelastic scattering”.
- The domain of validity of the theory should be defined.
What are the system requirements for the assumption of the tight binding theory to be valid?
What are the system requirement for the description of open systems with non hermitian Hamiltonians to be valid?
Split gate can now vary the confinement of Coulomb islands from completely isolated quantum dots to completely open 2D electron systems. In between, somewhere, are the open quantum systems described by this theory. It is therefore important to give a theoretical definition of this “somewhere”. Experimental situations where coupling might be too strong for tight binding theory to be inapplicable are frequently encountered in molecular electronics.
Minor comments
- After Eq.6: replace “identical” with “identity” matrix
- Above Eq.10: “which is diagonal in the momentum representation…”
Do you mean the Heisenberg representation ?
- Top of p.5: Description of Eq.19 and Eq.20 is a little confusing as the explanation refers back to the denominator of Eq.12 instead of Eq.18. It would be better to keep the description consistent with the same notation.
- line 123: “Moreover… near EP” is one of the most difficult sentences to follow.
- p15, line 220: “III-V semiconductor heterostructures” might be more widely understood than "A3B5 semiconductor heterostructures".
